# Stability and Predictors of Poor 6-min Walking Test Performance over 2 Years in Patients with COPD

**DOI:** 10.3390/jcm9041155

**Published:** 2020-04-18

**Authors:** Mª Piedad Sánchez-Martínez, Roberto Bernabeu-Mora, Mariano Martínez-González, Mariano Gacto-Sánchez, Rodrigo Martín San Agustín, Francesc Medina-Mirapeix

**Affiliations:** 1Department of Physical Therapy, University of Murcia, 30100 Murcia, Spain; mariapiedad.sanchez1@um.es (M.P.S.-M.); mamargon@um.es (M.M.-G.); mirapeix@um.es (F.M.-M.); 2Department of Pneumology, Hospital General Universitario J M Morales Meseguer, 30008 Murcia, Spain; 3Department of Physical Therapy, University of Girona, 17190 Girona, Spain; marianogacto@hotmail.com; 4Department of Physical Therapy, University of Valencia, 46010 Valencia, Spain; rodrigo.martin@uv.es

**Keywords:** 6MWT, COPD, predictors, 5STS, mobility activities

## Abstract

Poor performance in the 6-min walk test (6MWT < 350 m) is an important prognostic indicator of mortality and risk of exacerbations in patients with chronic obstructive pulmonary disease (COPD). Little is known about the stability of this state over time and what factors might predict a poor 6MWT performance. To determine the stability of 6MWT performance over a 2-year period in COPD patients participating in annual medical follow-up visits, and to assess the ability of several clinical, pulmonary, and non-pulmonary factors to predict poor 6MWT performance, we prospectively included 137 patients with stable COPD (mean age, 66.9 ± 8.3 years). The 6MWT was scored at baseline and 2-year follow-up. To evaluate clinical, pulmonary, and non-pulmonary variables as potential predictors of poor 6MWT performance, we used multiple logistic regression models adjusted for age, sex, weight, height, and 6MWT performance at baseline. Poor 6MWT performance was stable over 2 years for 67.4% of patients. Predictors of poor 6MWT performance included a five-repetition sit-to-stand test score ≤2 (OR, 3.01; 95% CI, 1.22–7.42), the percentage of mobility activities with limitations (OR, 1.03; 95% CI, 1.00–1.07), and poor 6MWT performance at baseline (OR, 4.64; 95% CI, 1.88–11.43). Poor 6MWT performance status was stable for the majority of COPD patients. Lower scores on the five-repetition sit-to-stand test and a higher number of mobility activities with limitations were relevant predictors of poor 6MWT performance over 2 years. Prognostic models based on these non-pulmonary factors can provide non-inferior discriminative ability in comparison with prognostic models based on only pulmonary factors.

## 1. Introduction

The 6-min walk test (6MWT) is the most popular test to measure exercise tolerance in patients with chronic obstructive pulmonary disease (COPD) [1,2]. It has proven to be reliable, inexpensive, safe and, easy to apply [3,4,5]. Impaired exercise tolerance is an important indicator of severity in COPD [6] and has previously been associated with decreased functional capacity to perform activities of daily living, as well as with a reduction in the quality of life [7,8]. Moreover, poor 6MWT performance (walking distances < 350 m) has been associated with a significant increase in mortality and a high risk of exacerbation [9,10]. 

Given the importance of impaired exercise tolerance and poor 6MWT performance (<350 m), several studies have analyzed the evolution of 6MWT over time [2,10], as well as pulmonary and non-pulmonary (e.g., peripheral muscle strength) cross-sectional associated factors [11,12]. Other studies have determined that the prevalence of poor 6MWT performance (<350 m) is over 41.0% among patients with COPD [11]. 

Despite the frequency of poor 6MWT performance and its negative impact on patients with COPD, little is known about the stability of this status over time and what factors could be used to predict poor 6MWT performance. 

Some cross-sectional studies have provided evidence that pulmonary factors such as dyspnea, health status, and the presence of depressive symptoms are determinants of poor 6MWT performance [11,13]. Nevertheless, their relevance to stability over time could differ widely. The relevance of non-pulmonary factors (e.g., peripheral muscle strength or physical performance) also remains unknown. Quadriceps strength has been shown to be related to the total distance of the 6MWT in meters but not with poor 6MWT performance [12]. Knowledge of the prognostic ability of some of these modifiable factors might be useful in developing interventions aimed at reducing poor 6MWT performance and improving patients’ ability to cope with the consequences of COPD [13]. 

The aim of this study was to describe the stability of 6MWT performance over a 2-year period and to test several non-pulmonary factors adjusted by clinical and pulmonary factors for their ability to predict poor 6MWT performance in stable COPD patients participating in annual medical follow-up visits. We also compared the temporal evolution of these prognostic non-pulmonary factors between patients with and without stable poor 6MWT performance over 2 years.

## 2. Materials and Methods

### 2.1. Study Design and Participants 

In this longitudinal study, patients with stable COPD were prospectively recruited from an outpatient pulmonary service at Morales Meseguer Hospital in Murcia, Spain, during 2015. All study participants provided written informed consent, and the study protocol was approved by the Institutional Review Board of the hospital (the Ethics Committee of Clinical Research of the General University Hospital; approval number EST-35/13). Patients were included in this study when they had a diagnosis of COPD, according to the Global Initiative for Chronic Obstructive Lung Disease (GOLD) recommendations (i.e., a post-bronchodilator ratio of forced expiratory volume in 1 s (FEV_1_)/forced vital capacity post-bronchodilator ratio of <70%) [14]; were in a stable state (without exacerbations in the previous 6 weeks); and were aged between 40 and 80 years. Patients were excluded when they displayed an unstable cardiac condition within 4 months of the start of the study, cognitive deterioration, or an inability to walk. Over a 1-year period, a consecutive sample of eligible patients was identified on a rolling basis, based on patient health examinations. A pulmonary physician assessed eligibility for inclusion among all patients with stable COPD who attended medical follow-up visits. The initial cohort included 137 patients. All of them participated in an annual follow-up program for 2 years, consisting of medical consultations including examinations and updating of their medical treatments; none of the participants were involved in a rehabilitation program. 

### 2.2. Measurements

Study data were obtained in patient interviews conducted at baseline (T1) and at 2 years (T2) after study initiation. At these two times, data on 6MWT performance and clinical, pulmonary, and non-pulmonary variables were obtained, but only baseline data were used for analysis as possible predictors of poor 6MWT performance. Patient interviews were conducted by a pulmonologist. For follow-up visits, research staff contacted patients 1–2 weeks before each follow-up time to ask them if their condition was stable. We required that the interview was conducted within 4 weeks of the due date. Subjects that were not interviewed within this time interval were removed from the study at that time point.

### 2.3. Outcome Measure

The 6MWT was performed indoors, along a flat, straight, 30-m walking course, supervised by two well-trained nurses (with a mean of 19 years of experience), according to the American Thoracic Society guidelines [4]. Patients were instructed and encouraged to walk as far as possible in 6 min, using standard incentive phrases every minute. Patients were permitted to stop and rest during the test, but were instructed to resume walking as soon as they felt able to do so. We used this test to classify patients with poor (<350 m) or non-poor (≥350 m) 6MWT performance at 2 years. Moreover, both baseline status, poor or non-poor 6MWT performance, were categorized as stable (when the status did not change from T1 to T2) or unstable (when it either improved or worsened). 

### 2.4. Predictor Variables 

We selected clinical and pulmonary variables of poor 6MWT performance as determined by Spruit et al., and selected the non-pulmonary factors based on their association with COPD [11,15]. These variables were classified into three domains: sociodemographic, clinical and pulmonary, and non-pulmonary.

The sociodemographic variables included age (years) and sex. Clinical and pulmonary variables included the body mass index (BMI, kg/m^2^) obtained by measuring the weight (kg) and height (m); dyspnea, measured with the self-administered modified British Medical Research Council (mMRC) [16]; perceived health status assessed by the COPD Assessment Test (CAT) [17]; and depression, assessed by the depression subscale of the Hospital Anxiety and Depression Scale (HADS), with probable depression considered when the total score was ≥11 [18]. Pulmonary function was assessed with spirometry, performed with a Master Scope Spirometer (version 4.6; Jaeger, Würzburg, Germany), according to the American Thoracic Society guidelines [19]. The patient’s medical history was reviewed to determine smoking pack-years, current smoker status, the number of comorbidities measured using the Functional Comorbidity Index [20], the number of exacerbations (“moderate” defined as use of corticosteroids and/or antibiotics; “severe” defined as requiring hospitalization), as well as history of hospitalization for exacerbations in the previous year, and the presence of heart disease.

The non-pulmonary variables included handgrip, quadriceps, and elbow strength that were assessed on the dominant side with a handgrip dynamometer (KERN MAP 80K1, KERN & Sohn GmbH 1, Balingen, Germany; for handgrip) and hand-held dynamometer (HHD) (Nicholas Manual Muscle Tester, model 01160, Lafayette Instrument Company, Lafayette, Indiana, USA; for quadriceps and elbow) [21]. To measure quadriceps strength, participants stayed seated with knee flexed 70°. The HHD was placed against the anterior aspect of the tibia, 5 cm above the lateral malleolus, and a break test was performed. For elbow and handgrip strength, participants stayed seated with their shoulder adducted 0°, elbow flexed 90°, and forearm in a neutral position [21]. The short physical performance battery (SPPB) was performed in the following sequence: (a) standing balance tests, (b) 4-m gait speed test (4MGS), and (c) five-repetition sit-to-stand motion test (5STS). The 4MGS test measures the time taken to walk 4-m within a clinical assessment room at usual speed. Timing begins when the participant first starts moving and stop when the participant’s first foot completely crosses the 4-m line. A summary score integrated the three performance measures and ranged from 0 to 12 [22]. The 5STS test required participants to rise from a chair with their arms across their chest, and then sit back down, five times. The 4MGS and 5STS test scores ranged from 1 to 4; scores <3 points on the 4MGS and <2 points on the 5STS indicated poor performance [23]. The self-reported mobility questionnaire used by Sternfeld comprised ten items related to mobility activity domains defined by the International Classification of Functioning, Disability, and Health (ICF) [24,25]. The patients assessed the degree of difficulty in performing the activity and the proportion of those activities with a limitation was calculated.

### 2.5. Statistical Analysis 

We used descriptive statistics to summarize baseline characteristics of the whole sample as the mean and standard deviation (SD) for continuous variables, and numbers and percentages for categorical data. We used the Pearson chi-square test and the independent *t*-test to examine between-group differences in criteria with respect to poor or non-poor 6MWT performances. 

Multivariate logistic regression was used to assess the predictive ability of several pulmonary and non-pulmonary factors on poor 6MWT performance (<350 m) at 2 years. In the first stage of the analysis, several models were constructed to explore clinical and pulmonary factors. First, model 1 was constructed using the enter method and included each clinical and pulmonary variable, adjusted by age, sex, weight, height, and baseline status of 6MWT performance (poor or non-poor). Second, a final model was constructed using a stepwise method including as candidate predictors all significant (*p* < 0.10) clinical and pulmonary variables from model 1 plus age, sex, weight, height, and 6MWT performance at baseline. 

In the second stage, several models were constructed to explore non-pulmonary predictors and their additional relevance to prognostic pulmonary factors. First, model 1 was constructed using the enter method and included each non-pulmonary variable, adjusted by age, sex, weight, height, and 6MWT performance at baseline. Second, model 2 included each non-pulmonary variable adjusted by significance (*p* < 0.10) from the final model regarding clinical and pulmonary variables. Third, a final non-pulmonary model was constructed using a stepwise method that included as candidate predictors all significant variables from model 2, clinical and pulmonary significant variables from the final model, and 6MWT performance at baseline. Goodness-of-fit and regression diagnostics for the models were assessed using methods described elsewhere [26]. 

We constructed receiver operating characteristic (ROC) curves, with area under the curve (AUC) data used to determine and compare the discriminative accuracy of both the final pulmonary model and the final non-pulmonary model on poor 6MWT performance. In accordance with previous authors, an AUC > 0.7 was used as the criterion of good discrimination [26]. 

To explore the temporal evolution of non-pulmonary predictors, we calculated their scores at T1 and T2, and the mean change between these points. We used *t*-tests to compare differences in these means in patients with stable versus unstable COPD over time, as well as in patients with poor versus non-poor 6MWT performance at baseline. 

Sample size calculation was based on the rule of thumb that 15 subjects per predictor are needed for a reliable equation [27]. We recruited a minimum of 90 participants assuming a maximum of 6 predictors. All analyses were performed using the Statistical Package for the Social Sciences (SPSS) version 24.0 (IBM SPSS, Chicago, IL, USA).

## 3. Results

### 3.1. Participants

At baseline, we included a total of 137 patients with a mean age of 66.9 years (87.6% were males and 29.9% still current smokers). These 55 (40.1%) subjects had poor 6MWT performance, with greater smoking pack-years and more dyspnea, exacerbations, and hospitalizations for COPD and a lower percentage of FEV_1_. There were also differences in all non-pulmonary variables in comparison with subjects with non-poor 6MWT performance (Table 1). 

During follow-up, 119 (86.9%) patients remained at T2. Of those lost to follow-up, 6 (4.3%) died, 8 (5.8%) dropped out due to lung cancer, and 4 (2.9%) chose not to continue. No patients were removed due to an unstable stage or an exacerbation that occurred close to any follow-up visit. The patients lost during follow-up were not significantly different from those who continued throughout the study.

### 3.2. Stability and Predictors of Poor 6MWT Performance over a 2-Year Period

At 2 years, 29 (67.4%) of the 55 patients who had poor 6MWT performance at baseline were stable, remaining at the same status, and 14 (32.66%) improved. In similar percentages, 56 (74.3%) of the 82 patients who started with a non-poor 6MWT performance were stable and 19 (25.3%) worsened. Consequently, the prevalence of poor 6MWT performance at T1 and T2 were very similar (40.1% vs. 40.3%, respectively) (Figure 1).

Results of the multivariate models for clinical and pulmonary predictors of poor 6MWT performance are shown in Table 2. Model 1 shows that poor 6MWT performance was positively associated with the presence of heart disease and depression, dyspnea ≥ 2, CAT ≥ 10, and GOLD stage D, and was negatively associated with a higher percentage FEV_1_ and SpO_2_, even after adjustment for other relevant clinical determinants. The final model, which included all significant clinical and pulmonary variables from model 1 plus sociodemographic covariates and 6MWT performance at baseline, showed that participants with heart disease, depression, baseline poor 6MWT performance, and lower percentage of SpO_2_ had a greater probability of poor 6MWT performance at 2 years. 

Table 3 shows the non-pulmonary predictors that could predict poor 6MWT performance at 2 years. Model 1, which included each non-pulmonary factor adjusted for sociodemographic covariates and 6MWT performance at baseline, and model 2, which included all significant variables from the final model of Table 2 and 6MWT performance at baseline, showed that lower quadriceps strength, 5STS test score ≤ 2, and higher percentages of mobility activities with limitations were candidate predictors of poor 6MWT performance in COPD patients. Nevertheless, in the final model only 5STS test score ≤ 2, a higher percentage of mobility activities with limitations, and 6MWT performance at baseline remained as independent predictors of poor 6MWT performance over 2 years. No pulmonary factors were included. 

Figure 2 shows the ROC plots and the area under the ROC curve of two final models from Table 2 and Table 3 for predicting poor 6MWT performance over 2 years. The pulmonary model included heart disease, depression, and percentage of SpO_2_ (Table 2), and the non-pulmonary model included the 5STS test and the percentage of mobility activities with limitations (Table 3). The two were adjusted by baseline status (poor or non-poor 6MWT performance). The AUCs of the two models were similar (0.811 vs. 0.806), implying that non-pulmonary factors have non-inferior discriminative ability compared to pulmonary factors.

### 3.3. Temporal Evolution of Relevant Non-Pulmonary Predictors over 2 Years

Figure 3 shows the means of the 5STS values (in seconds) and the mobility activities with limitations (as percentages) at T1 and T2 for both stable and unstable patients who had poor 6MWT performance at baseline. Both stable and unstable patients showed similar temporal evolution concerning the percentage of mobility activities with limitation, their mean changes being 2.29% and 2.61%, respectively (Table 4). In contrast, the 5STS time evolved slightly differently in the stable and unstable patients. Nevertheless, Table 4 shows that the differences between the changes in their mean times were not statistically significant (*p* = 0.199). Similarly, there were also no significant differences in 5STS and mobility activities between stable and unstable patients who had non-poor 6MWT performance at baseline.

## 4. Discussion

In the present study, we primarily examined the stability of 6MWT performance over a 2-year period and evaluated predictors of poor 6MWT performance in stable COPD patients who participated in annual medical follow-up visits. Poor 6MWT performance was relatively unstable over time for one-third of patients. This instability was shown among patients from both groups of poor and non-poor 6MWT performance at baseline. In spite of this instability, the results revealed that the prevalence of poor 6MWT performance over time was similar at baseline and at the end of 2 years, because non-poor 6MWT performance was also relatively unstable for one-quarter of the patients. We found that the most relevant predictors of poor 6MWT performance were the 5STS test, the percentage of mobility activities with limitation, and the 6MWT performance status at baseline.

To our knowledge, this is the first study to describe the stability of poor 6MWT performance using the cut-point of <350 m in patients with COPD. Our results showed that poor 6MWT performance was stable over 2 years for two-thirds of patients. Previous studies that explored the temporal mean change in 6WMT in meters also found that this test was stable over time for the majority of the population with COPD, except for patients with severe airflow obstruction (GOLD stage D), who showed greater decline [2,10,28]. Our study also provides evidence that GOLD stage D patients have a greater predisposition to poor 6MWT performance over time. This could reflect the deconditioning caused by a more sedentary lifestyle in patients with a higher level of disease severity [2,10,28]. However, it is also possible that the decrease results from the development of a more advanced systemic involvement of COPD [29].

Our study showed that several clinical and pulmonary factors were cross-sectionally associated with poor 6MWT performance at baseline. Patients who walked less than 350 m had greater smoking pack-years, dyspnea ≥2, a lower percentage of FEV_1_, and a higher number of hospitalizations for exacerbations. These results are consistent with the study by Spruit et al., which examined the determinants of poor 6MWT performance transversally [11]. Similarly, Waatevik et al. found that these characteristics were related to a shorter distance walked in the 6MWT [30]. Nevertheless, our study found that these factors were not longitudinally associated. Only the presence of cardiac pathology and depression and the percentage of SpO_2_ were relevant as predictors of poor 6MWT performance over time. The final model including these three factors provided very good discriminative accuracy for poor 6MWT performance, with an excellent AUC.

Our study also showed that two non-pulmonary factors were independent predictors of poor 6MWT performance: the 5STS test and the number of mobility activities with limitations. Moreover, our study evidenced that these factors had similar discriminative accuracy than some pulmonary factors, and even more predictive ability in the final non-pulmonary model. Previous authors have associated these factors with 6MWT, but used cross-sectional studies [23,28]. Kapella et al. found that the distance walked in the 6MWT test was related to lower performance as measured with the Functional Performance Inventory-Short Form (FPI-SF) questionnaire, which is an approximate measure of the deterioration of several activities [28]. Bernabeu-Mora et al. found that scores lower than 2 in the 5STS were useful in discriminating poor 6MWT performance in patients with COPD [23]. The relevance of the 5STS test could be due to the extra respiratory demand required in this sit-to-stand test [31].

Regarding the 2-year temporal evolution of the mean scores of non-pulmonary factors, our results showed that patients with stable and with unstable 6MWT performance showed a similar evolution of 5STS test results and limited mobility activities. This finding was expected because Kapella et al., who also analyzed stable COPD patients participating in medical follow-up visits, found that mobility-activity functioning did not change significantly over 3 years [28]. Nevertheless, in spite of this lack of change, for us it was an unexpected result that 33% of patients with poor 6MWT performance at baseline improved over 2 years. In our opinion, this finding suggests that other concurrent changes could be responsible for the instability in that 33% of cases.

### 4.1. Implications for Practice and Research

It seems reasonable that the presence of poor 6MWT is multifactorial, and that non-pulmonary outcomes such as mobility activities and the 5STS test should be considered prognostic factors of this status in patients with stable COPD [11]. Moreover, our study showed that non-pulmonary factors can provide non-inferior discriminative ability in comparison with pulmonary factors. It suggests that both models are valid for use in daily clinical practice, and that when time and space are limited, one of the two models may be chosen.

In spite of suggestions in the existing literature that professional interventions can modify 5STS test results and mobility activity limitations [32,33], our study found no change in the scores of these main non-pulmonary predictive factors over 2 years. This suggests that patient participation in annual medical follow-up visit programs does not change these scores by itself. Evidence-based thinking suggests that the addition of a rehabilitation program could improve them [32,33]. Nevertheless, more research is needed to know whether improvement in these factors could lead to reduced prevalence of poor 6MWT performance in patients with COPD.

### 4.2. Strengths and Limitations of the Study

First, this was a longitudinal study. The main advantage of a longitudinal analysis is that changes can be analyzed in individual patients. In addition, it allows measurement of the prevalence of 6MWT performance at more than one moment in time. Our evaluation of the data from the two combined time points may also have provided a more realistic estimate of the prevalence, which may have strengthened the results. Second, our models included a wide variety of previously identified pulmonary factors of poor 6MWT performance and many non-pulmonary factors which had not previously been longitudinally analyzed in relation to poor 6MWT performance.

Our study also has several limitations. First, although we included a wide variety of possible predictive factors of poor 6MWT performance in our model, there could be other factors not included in our model that could have improved its predictive power (e.g., the quantity of drugs taken regularly, participation in rehabilitation programs, etc.) [33,34]. Second, although measurements were made at two time points (once a year) to determine the stability of 6MWT performance over time, it is likely that additional changes occurred between these time points. Finally, due to the small number of women in the cohort, the results in women must be interpreted with caution, because they might not be generalizable to all women with COPD.

## 5. Conclusions

Our results demonstrate that 6MWT performance status was stable for the majority of stable COPD patients, but a third of patients showed instability, changing either from non-poor to poor status or vice versa. These results suggested that the nature of 6MWT performance was dynamic for a part of the patient population with COPD. We also showed that lower scores in the 5STS test and a higher number of mobility activities with limitations were relevant predictors of poor 6MWT performance, and that prognostic models based on these non-pulmonary factors can provide non-inferior discriminative ability in comparison with prognostic models based only on pulmonary factors. Finally, our study suggests that patient participation in annual medical follow-up visit programs by itself does not cause changes in those non-pulmonary factors.

## Figures and Tables

**Figure 1 jcm-09-01155-f001:**
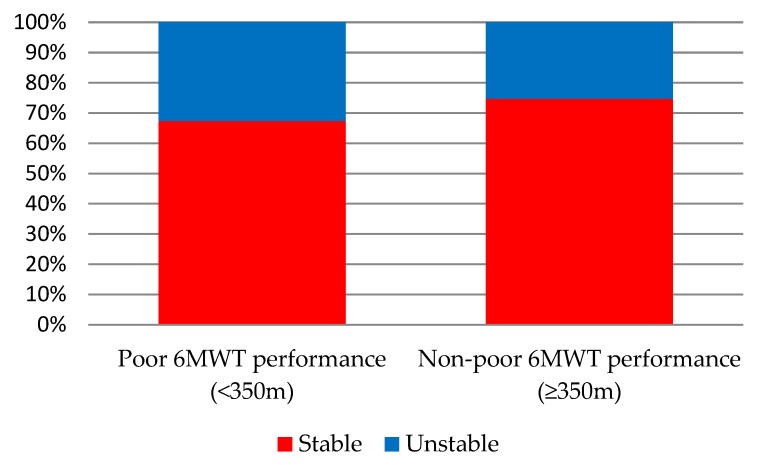
Stability of 6-min walk test (6MWT) performance after 2 years, according to baseline status.

**Figure 2 jcm-09-01155-f002:**
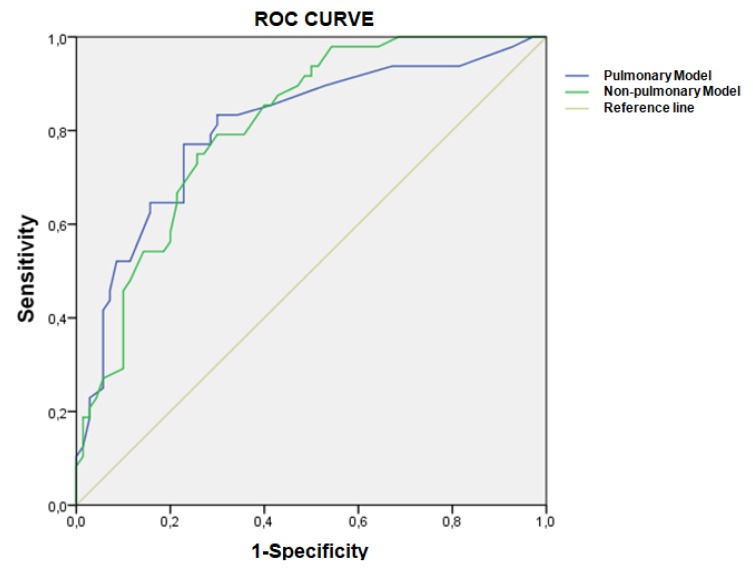
Receiver operating characteristic curves comparing the final clinical pulmonary model and final non-pulmonary model as predictors of poor 6MWT performance (<350 m).

**Figure 3 jcm-09-01155-f003:**
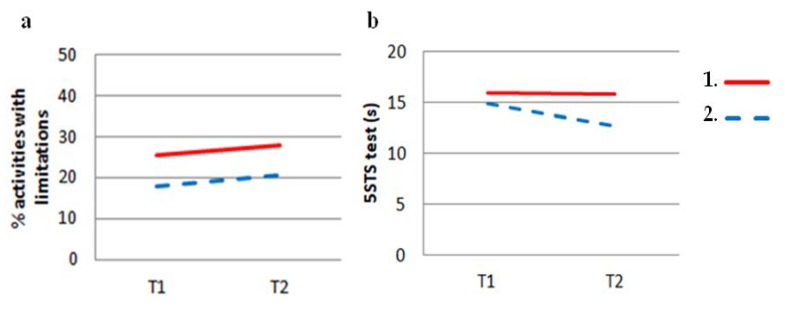
Mean of mobility activities with limitations (**a**) and 5STS test times (**b**) at T1 and T2 between stable (1) and unstable (2) patients who had poor 6MWT performance at baseline.

**Table 1 jcm-09-01155-t001:** Characteristics of the study population (*n* = 137), stratified according to poor 6MWT performance ^a^.

Characteristics	All (*n* = 137)	<350 m (*n* = 55)	≥350 m (*n* = 82)	*p*-Value
**Sociodemographic variables**				
Age (years), mean ± SD	66.9 ± 8.3	69.97.8	64.8 ± 8.0	0.000 *
Male	120(87.6)	51(92.7)	69(84.1)	0.135
**Clinical and Pulmonary variables**				
BMI (kg/m^2^), mean ± SD	28.9 ± 5.0	28.6 ± 5.4	29.0 ± 4.8	0.645
Smoking, mean pack-years ±SD	58.7 ± 25.5	66.7 ± 26.1	53.4 ± 23.7	0.002 *
Current smoker	10(29.9)	14(25.5)	27(32.9)	0.349
Number Comorbidities, mean ± SD	3.1 ± 1.6	3.1 ± 1.6)	3.1 ± 1.7	0.831
Heart disease (yes)	19(13.9)	10(18.2)	9(11.0)	0.232
Dyspnea (mMRC ≥ 2)	49(35.2)	29(52.7)	20(24.4)	0.001 *
Number of Exacerbations ^b^, mean ± SD	1.4 ± 1.4)	2.3 ± 1.9	1.7 ± 1.4	0.034 *
CAT ≥ 10	100(73.0)	43(78.2)	57(69.5)	0.263
FEV_1_ (litres), mean ± SD	1.3 ± 0.4	1.1 ± 0.4	1.4 ± 0.4	0.001 *
FEV_1_ (% predicted), mean ± SD	50.2 ± 16.5	46.0 ± 16.5	53.0 ± 16.0	0.014 *
FVC (litres), mean ± SD	2.2 ± 0.6	1.9 ± 0.5	2.3 ± 0.7	0.001 *
FVC (% predicted), mean ± SD	66.6 ± 18.7	61.5 ± 17.3	70.1 ± 18.9	0.008 *
FEV_1_/FVC ratio, mean ± SD	57.8 ± 8.2	56.6 ± 9.0	58.6 ± 7.5	0.144
GOLD stage				0.270
A	24(17.5)	9(16.4)	15(18.3)	
B	22(16.1)	8(14.5)	14(17.1)	
C	12(8.8)	2(3.6)	10(12.2)	
D	79(57.7)	36(65.5)	43(52.4)	
SpO_2_, mean ± SD	94.4 ± 0.6	94.0 ± 2.0	94.7 ± 2.2	0.091
Oxygen Therapy (Yes)	7(5.1)	5(9.1)	2(2.4)	0.083
History of Hospitalized Exacerbation (Yes)	85(62.0)	43(78.2)	42(51.2)	0.001 *
Depression (HAD-D ≥ 11)	13(9.5)	7(12.7)	6(7.3)	0.290
**Non-pulmonary variables**				
Handgrip strength (kg), mean ± SD	28.3 ± 8.0	27.5 ± 7.4	28.8 ± 8.3	0.327
Quadriceps strength (kg), mean ± SD	15.7 ± 2.8	14.8 ± 2.9	16.3 ± 2.7	0.003 *
Elbow strength (kg), mean ± SD	15.0 ± 3.2	13.9 ± 2.4	15.7 ± 3.5	0.001 *
6-min walk test (meters), mean ± SD	349.1 ± 84.7	271.3 ± 69.8	403.3 ± 40.1	0.001 *
SPPB score, mean ± SD	9.5 ± 2.0	8.6 ± 1.9	10.1 ± 1.8	0.001 *
5STS ≤ 2	75(54.7)	42(76.4)	33(40.2)	0.001 *
4MGS ≤ 3	45(32.8)	29(52.7)	16(19.5)	0.001 *
% activities with limitations	21.6(17.5)	3.8(1.9)	1.5(1.3)	0.001 *

Abbreviations: SD, standard deviation; BMI, body mass index; mMRC, modified British Medical Research Council; CAT, COPD assessment test; FEV_1_, forced expiratory volume in 1 s; GOLD, Global Initiative for Chronic Obstructive Pulmonary Disease; HAD, Hospital Anxiety and Depression Scale; BODE, body mass index, airflow obstruction, dyspnea, and exercise capacity; SPPB, short physical performance battery; 5STS, five-repetition sit-to-stand test; 4MGS, 4-m gait-speed. ^a^ Values represent the number (%) of participants in each group, unless otherwise noted. ^b^ Moderate or severe exacerbations in the previous year. * *p* < 0.05.

**Table 2 jcm-09-01155-t002:** Multivariate regression models show the predictive strengths of clinical and pulmonary factors for poor 6MWT performance (<350 m) ^a^.

Predictors	Model 1 ^b^	Final Model ^c^
Smoking, pack-years	1.00(0.99–1.02)0.323	
Current smoker	1.35(0.54–3.41)0.517	
BMI	0.86(0.48–1.55)0.622	
Number of comorbidities	1.07(0.79–1.45)0.626	
Heart disease (yes)	5.21(1.29–21.06)0.020 *	6.07(1.56–23.49)0.009 *
SpO_2_	0.80(0.64–1.01)0.063 ^t^	0.80(0.64–1.00)0.054 ^t^
Dyspnea(mMRC ≥ 2)	2.51(1.09–6.34)0.048 *	
Exacerbations ≥2	1.60(0.67–3.80)0.288	
CAT ≥ 10	3.88(1.37–10.93)0.010 *	
FEV_1_ (% of predicted)	0.97(0.94–0.99)0.036 *	
GOLD	0.010	
A	Ref.	
B	2.07(0.39–10.80)0.386	
C	1.07(0.13–8.55)0.947	
D	6.68(1.76–25.39)0.005 *	
History of Hospitalized Exacerbation (Yes)	1.39(0.58–3.33)0.460	
Depression (HAD-D ≥ 11)	3.13(0.93–10.46)0.063 ^t^	3.87(1.19–12.49)0.024 *

Abbreviations: BMI, body mass index; mMRC, modified British Medical Research; CAT, COPD assessment test; FEV_1_, forced expiratory volume in 1s; GOLD, Global Initiative for Chronic Obstructive Pulmonary Disease; HAD, Hospital Anxiety and Depression Scale. ^a^ Values are presented as odds ratio (95% confidence interval) and *p*-value. ^b^ Model 1 includes each predictor adjusted for age, sex, weight, height, and 6MWT performance at baseline. ^c^ The final model is fully adjusted for significant variables from model 1 and age, sex, weight, height, and 6MWT performance at baseline. * *p* < 0.05. ^t^
*p* < 0.10.

**Table 3 jcm-09-01155-t003:** Multivariate regression models show the predictive strengths of non-pulmonary factors for poor 6MWT performance (<350 m) ^a^.

Predictors	Model 1 ^b^	Model 2 ^c^	Final Model ^d^
Handgrip strength (kg)	0.96(0.88–1.04)0.369	0.97(0.91–1.03)0.359	
Quadriceps strength (kg)	0.83(0.70–0.99)0.048 *	0.85(0.71–1.01)0.073 ^t^	
Elbow strength (kg)	1.04(0.86–1.24)0.661	1.00(0.85–1.17)0.963	
SPPB (range 0 ± 12)	0.77(0.58–1.03)0.082	0.79(0.59–1.06)0.123	
5STS ≤ 2	3.88(1.51–10.00)0.005 *	2.80(1.10–7.10)0.030 *	3.01(1.22–7.42)0.016 *
4MGS ≤ 3	1.60(0.60–4.26)0.338	1.85(0.68–5.06)0.226	
Self-reported questionnaire (0 ± 100)	1.04(1.00–1.07)0.011 *	1.02(0.99–1.06)0.010 *	1.03(1.00–1.07)0.015 *

Abbreviations: SPPB, short physical performance battery; 5STS, five-repetition sit-to-stand test; 4MGS, 4-m gait speed. ^a^ Values are presented as odds ratio (95% confidence interval) and *p* value. ^b^ Model 1 includes each predictor adjusted for age, sex, weight, height, and 6MWT performance at baseline. ^c^ Model 2 is fully adjusted for significant variables from the final model in Table 2. ^d^ The final model includes all significant variables from model 2 and 6MWT performance at baseline. * *p* < 0.05. ^t^
*p* < 0.10.

**Table 4 jcm-09-01155-t004:** Mean change between T1 and T2 in the five-repetition sit-to-stand motion test (5STS) and mobility activities with limitations between stable and unstable patients who had poor and non-poor 6MWT performance status at baseline.

	<350 m at Baseline			≥350 m at Baseline		
Predictors	Stable at 2 Years (*n* = 56)	Unstable (Improvement) at 2 Years (*n* = 14)	*p*-Value	Stable at 2 Years (*n* = 29)	Unstable (Worsening) at 2 Years (*n* = 19)	*p*-Value
% activities with limitations	2.29(20.93)	2.61(22.98)	0.964	−2.55(10.99)	7.01(20.05)	0.060
5STS (s)	−0.22(3.12)	−1.51(1.79)	0.199	−1.10(2.15)	−0.33(3.16)	0.269

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
