# Peer review of "Stability and Predictors of Poor 6-min Walking Test Performance over 2 Years in Patients with COPD"

_jcm, 2020, doi:10.3390/jcm9041155_

Round 1
Reviewer 1 Report
- The authors have undertaken what seems to have been a carefully done study on 6MWTs in patients with COPD, and then divided them into poor and not so poor (< or > 350m), then investigated baseline predictors of this dichotomous divide. They also investigated change in category over 1 year and predictors for this, although not to the same depth (no ROCs for example, and an explanation for that should be given). They developed separate models for clinical-pulmonary and non-pulomonary putative predictors (although oddly BODE index was included in this which seems anomalous. The outcomes are of interest, though for the most part not surprising as they largely represented severity of COPD as well as those older, fatter and sicker. There seemed to be some independent predictive value for 5STS overall, but not for changes over 1 year.
- The statistical analysis I found rather laboured, messy and confusing. Some analyses were done for baseline values and some for values "over 2 years" but I did not get what that meant I`m afraid. The dichotomous division of 6MWT may be of clinical utility, but in this sort of contributory risk analysis, then I would much have preferred 6MWT to be used as a continuous variable, both for baseline values and change with time. It might also have been worth a rather more sophisticated approach by dividing factors into those likely to be directly reflective of lung disease and its causes(lung function, and probably smoking, for example), then potential confounders such as heart disease, obesity and other co-morbidities, and then potential pulmonary and non-pulmonary mediators of poor 6MWT such as symptoms and muscle strength/activity. Depression may be in the latter box, but may also just be more an outcome of poor 6MWT and so related but an effect rather than on causal pathway. There is good literature on doing this well, and good software to help.
- I presume the cross-overs from one clinical division to another would have been in those close to the break point of </>350m; it would be interesting to know wht rnge they were in, and I would suggest another reason for looking at the data as continuous.
- There was no information given on a number of things that could have been influential e.g. actual time since last acute exacerbation (AE) an number of AEs in the 12 months before recruitment; treatments , as both ICS and LABAs could affect upper leg strength; and more sophisticated lung function such as RV (to indicate air trapping), DLco to indicate emphysema. Sensitive CRP could also indicate chronic inflammation. These issues ought t least to be discussed.
Lesser issues
5. I was interested that heart disease was positive in some analyses and mot others; why might that be?
6. As well as post-BD spirometry, pre-values with BDR may have been informative as the more asthmatic group might have performed differently.
7. In Table 1, for the most part only 1 decimal place would be sufficient. Table 3 has some alignment issues. Some attention in tables is needed on ensuring units and % where used are stated.
8. The nature of the 4MGS test needs more explanation. In Results, it might be worth ensuring that with data some indiction of whether values or changes are good or bad in direction.
9. Discussion Para1, because 30% chnge in each direction, given the same numbers at baseline as at one year, is not an indiction of stability. I would be more discerning than this.
10. There are some odd typos, so a good re-read and edit is indicated.
Author Response
Response to Reviewer 1 Comments
We want to thank reviewer 1 for providing us with suggestions to improve our manuscript.
Point 1. Reviewer’s comment: The authors have undertaken what seems to have been a carefully done study on 6MWTs in patients with COPD, and then divided them into poor and not so poor (< or > 350m), then investigated baseline predictors of this dichotomous divide. They also investigated change in category over 1 year and predictors for this, although not to the same depth (no ROCs for example, and an explanation for that should be given). They developed separate models for clinical-pulmonary and non-pulmonary putative predictors (although oddly BODE index was included in this which seems anomalous). The outcomes are of interest, though for the most part not surprising as they largely represented severity of COPD as well as those older, fatter and sicker. There seemed to be some independent predictive value for 5STS overall, but not for changes over 1 year.
Response 1: Thank you very much for your suggestion. We only investigated change of poor 6-minute walk test (6MWT) performance over 2 years and predictors for this aspect. Our intention was to test several non-pulmonary factors adjusted by clinical and pulmonary factors. This aspect has been clarified within the text, line 57. For this purpose we did not compare ROC curves for isolated variables but, instead, we chose to test non-pulmonary factors adjusted by final models of clinical and pulmonary factors. On the other hand, we totally agree with your comment concerning the BODE index: subsequently, it has been removed from Table 1. Moreover, it was not used in the final analyses.
Point 2.1. Reviewer’s comment: The statistical analysis I found rather labored, messy and confusing. Some analyses were done for baseline values and some for values "over 2 years" but I did not get what that meant I`m afraid.
Response 2.1: We only use the baseline data on sociodemographic, non-pulmonary, and pulmonary variables as predictors of poor 6MWT performance. This aspect has been clarified in the text, lines 84-85.
Point 2.2. Reviewer’s comment: The dichotomous division of 6MWT may be of clinical utility, but in this sort of contributory risk analysis, then I would much have preferred 6MWT to be used as a continuous variable, both for baseline values and change with time. It might also have been worth a rather more sophisticated approach by dividing factors into those likely to be directly reflective of lung disease and its causes (lung function, and probably smoking, for example), then potential confounders such as heart disease, obesity and other co-morbidities, and then potential pulmonary and non-pulmonary mediators of poor 6MWT such as symptoms and muscle strength/activity. Depression may be in the latter box, but may also just be more an outcome of poor 6MWT and so related but an effect rather than on causal pathway. There is good literature on doing this well, and good software to help.
Response 2.2: Your suggestion is very interesting. On the one hand, we decided to use the dichotomous division of 6MWT since it is much more useful from a clinical perspective. Furthermore, the use of 6MWT as a continuous variable has been extensively studied in the literature, both longitudinally and transversally, as reflected in the introduction, and our intention was to corroborate that it occurs in the same way with the dichotomous division of this test. On the other hand, the sequence of analysis you propose is interesting, but it is not specific enough for our research purpose: to determine if the non-pulmonary factors (i.e. quadriceps strength, SPPB, etc,) adjusted by clinical and pulmonary factors could predict poor 6MWT performance at 2 years. We have clarified this aspect in the text, line 57.
Point 3. Reviewer’s comment: I presume the cross-overs from one clinical division to another would have been in those close to the break point of </>350m; it would be interesting to know what range they were in, and I would suggest another reason for looking at the data as continuous.
Response 3: We greatly appreciate your indication. We decided to use this cut-off point due to its clinical utility, since it is associated with an increase in mortality and exacerbations. Several studies have also used it. In the introduction section we clarified this aspect on lines 38-39 (“walking distances <350m has been associated with a significant increase in mortality and a high risk of exacerbation [1, 2].”)
Point 4. Reviewer’s comment: There was no information given on a number of things that could have been influential e.g. actual time since last acute exacerbation (AE) an number of AEs in the 12 months before recruitment; treatments, as both ICS and LABAs could affect upper leg strength; and more sophisticated lung function such as RV (to indicate air trapping), DLco to indicate emphysema. Sensitive CRP could also indicate chronic inflammation. These issues ought t least to be discussed.
Response 4: We collected data on the number of total exacerbations in the previous year, which includes moderate defined as “use of corticosteroids and/or antibiotics” and severe defined as “requiring hospitalization”. We also obtained the history of hospitalization for exacerbation in the previous 12 months. This aspect has been clarified within the text, line 116. On the other hand, we did not collect data on medical treatment, so we cannot analyze its relationship with 6MWT performance.
Lesser issues
Point 5. Reviewer’s comment: I was interested that heart disease was positive in some analyses and mot others; why might that be?
Response 5: In our study, the presence of heart disease was associated with having a greater probability of poor 6MWT performance at 2 years for clinical and pulmonary model. However, it is not relevant for non-pulmonary model. That could be due to the fact that non-pulmonary factors are more important for the 6MWT performance. In this sense, Spruit et al [3] did not find an association between heart disease and poor 6MWT performance cross-sectionally, in line with our findings.
Point 6. Reviewer’s comment: As well as post-BD spirometry, pre-values with BDR may have been informative as the more asthmatic group might have performed differently.
Response 6: Your suggestion is very interesting. This study included some patients with asthma and the mixed ACOS phenotype but only the 6MWT performance was analyzed for patients with COPD.
Point 7. Reviewer’s comment: In Table 1, for the most part only 1 decimal place would be sufficient. Table 3 has some alignment issues. Some attention in tables is needed on ensuring units and % where used are stated.
Response 7: According to your suggestions this aspect has been clarified in the text, Table 1 and Table 3.
Point 8. Reviewer’s comment: The nature of the 4MGS test needs more explanation. In Results, it might be worth ensuring that with data some indication of whether values or changes are good or bad in direction.
Response 8: We have clarified this aspect in the text, lines 127-129.
Point 9. Reviewer’s comment: Discussion Para1, because 30% change in each direction, given the same numbers at baseline as at one year, is not an indication of stability. I would be more discerning than this.
Response 9: According to your suggestions, we have highlighted the relevance of the instability of the 6MWT performance at the beginning of our discussion (lines 294-295) and conclusion section (lines 371-373).
Point 10. Reviewer’s comment: There are some odd typos, so a good re-read and edit is indicated.
Response 10: Thank you very much for your suggestion; we have reviewed the entire document.
Reviewer 2 Report
The paper deals with a very pragmatic topic. Authors developed a prospective study including COPD patients to assess the stability of 6MWT overt a 2 year period. The study design is clear, methods are adequately discussed . Tables are linear.
I invite the Authors to better explore the following points:
- There is a large distance between the proportion of patients with MMRC > 2 and CAT > 10. Conversely a recent paper published on ERJ showed a linear good interaction with kappa of agreement 0.74 https://erj.ersjournals.com/content/54/suppl_63/PA703 . Hoe the Authors explain this point?
- Please insert in the table absolute FEV1, FVC , FVC% and FEV/FVC
- The mean FEV1 is 50%. Any patients in long term oxygen therapy ? Please add information in this case.
- Number of comorbidities in se lacks of great specificity ; please use Charlson index which better describes the frailty.
- Please add some information about the nonrespiratory parameters measurement (handgrip , etc. )
- Whether Authors analysed carefully predictor of poor performance 6mwt , there is lack of data about the patients who initially performed more than 350 mt and then decline. Why patients worsened across the time?
- Do the Authors repeated the spirometry during the two years; in this case any association between the FEV1 decline and persistence of poor 6MWD ; any association between FEV1 decline and worsening in patients who initially achieved more than 350 mt.
Author Response
Response to Reviewer 2 Comments
We wish to thank reviewer 2 for the suggestions provided to improve our manuscript.
Point 1. Reviewer’s comment: There is a large distance between the proportion of patients with MMRC > 2 and CAT > 10. Conversely a recent paper published on ERJ showed a linear good interaction with kappa of agreement 0.74 https://erj.ersjournals.com/content/54/suppl_63/PA703. How the Authors explain this point?
Response 1: The difference in proportion of patients with mMRC≥2 and CAT ≥10 may be due to the type of population, since they are stable patients who have participated in annual medical follow-up visits and, therefore, may have a greater perception of deterioration in their quality of life than at a respiratory symptomatology level.
Point 2. Reviewer’s comment: Please insert in the table absolute FEV1, FVC, FVC% and FEV/FVC.
Response 2: This aspect has been inserted in Table 1.
Point 3. Reviewer’s comment: The mean FEV1 is 50%. Any patients in long term oxygen therapy? Please add information in this case.
Response 3: Only 7 patients had long term oxygen therapy. According to your suggestions this aspect has been added in Table 1.
Point 4. Reviewer’s comment: Number of comorbidities in se lacks of great specificity; please use Charlson index which better describes the frailty.
Response 4: We greatly appreciate your indication. We decided to use the Functional Comorbidity Index to measure the number of comorbidities because it is a reliable instrument and is related with the physical function. We have clarified this aspect in the text, line 114.
Point 5. Reviewer’s comment: Please add some information about the nonrespiratory parameters measurement (handgrip , etc.).
Response 5: According to your suggestions this aspect has been added in the text, lines 121-125 and lines 127-129.
Point 6. Reviewer’s comment: Whether Authors analysed carefully predictor of poor performance 6MWT, there is lack of data about the patients who initially performed more than 350 mt and then decline. Why patients worsened across the time?
Response 6: Lines 203 and 204 state that “of the 82 patients who started with a non-poor 6MWT performance 19 (25.3%) worsened”. These patients were also included in the multivariate analysis. All of these patients, and also those with poor 6MWT performance at baseline, were included in the analysis to explore the predictive capacity of non-pulmonary factors. Since 6MWT performance at baseline is a determinant of final status at 2 years, these analyses were adjusted by it.
Point 7. Reviewer’s comment: Do the Authors repeated the spirometry during the two years; in this case any association between the FEV1 decline and persistence of poor 6MWD ; any association between FEV1 decline and worsening in patients who initially achieved more than 350 mt.
Response 7: We only analyzed the temporal evolution of the poor 6MWT performance after 2 years (i.e. 5STS test and mobility activities with limitation). For this reason, the temporal relationship of FEV1 with 6MWT performance was not analyzed.
References
- Spruit MA, Polkey MI, Celli B, Edwards LD, Watkins ML, Pinto-Plata V, et al. Predicting outcomes from 6-minute walk distance in chronic obstructive pulmonary disease. J Am Med Dir Assoc. 2012 Mar;13(3):291-7.
- Pinto-Plata VM, Cote C, Cabral H, Taylor J, Celli BR. The 6-min walk distance: change over time and value as a predictor of survival in severe COPD. Eur Respir J. 2004 Jan;23(1):28-33.
- Spruit MA, Watkins ML, Edwards LD, Vestbo J, Calverley PM, Pinto-Plata V, et al. Determinants of poor 6-min walking distance in patients with COPD: the ECLIPSE cohort. Respir Med. 2010 Jun;104(6):849-57.
Round 2
Reviewer 1 Report
comments are for Editor only
Reviewer 2 Report
minor text editing, please check carefully.
For example:
Line 102 'based on'
Table 1 FEV1% All: Be careful when approximating 16.49 becomes 16.5 not 16.4; the same for other values (for instance SpO2).
Author Response
We wish to thank reviewer 2 for the suggestions provided to improve our manuscript.
Point 1. Reviewer’s comment: Table 1 FEV1% All: Be careful when approximating 16.49 becomes 16.5 not 16.4; the same for other values (for instance SpO2).
Response 1: We greatly appreciate your indication. This aspect has been modified in Table 1.